# Scalable Deep Kernel Gaussian Process for Vehicle Dynamics in Autonomous Racing

**Jingyun Ning**
Department of Electrical and Computer Engineering
University of Virginia United States
`jn2ne@virginia.edu`

**Madhur Behl**
Department of Computer Science
University of Virginia United States
`madhur.behl@virginia.edu`

**Abstract:** Autonomous racing presents a challenging environment for testing the limits of autonomous vehicle technology. Accurately modeling the vehicle dynamics (with all forces and tires) is critical for high-speed racing, but it remains a difficult task and requires an intricate balance between run-time computational demands and modeling complexity. Researchers have proposed utilizing learning-based methods such as Gaussian Processes (GP) for learning vehicle dynamics. However, current approaches often oversimplify the modeling process or apply strong assumptions, leading to unrealistic results that cannot translate to real-world settings. In this paper, we proposed DKL-SKIP, a method combining deep kernel learning (DKL) with SKIP-GP, for vehicle dynamics modeling. Our approach outperforms standard GP methods and the Numerical algorithms for Subspace State Space System Identification technique (N4SID) in terms of prediction accuracy. In addition to evaluating DKL-SKIP on real-world data, we also evaluate its performance using a high-fidelity autonomous racing AutoVerse simulator. The results highlight the potential of DKL-SKIP as a promising tool for modeling complex vehicle dynamics in both real-world and simulated environments.

**Keywords:** Gaussian Processes, Vehicle Dynamics, Autonomous Vehicle, Deep Kernel Learning

## 1 Introduction

The rising popularity of self-driving cars has inspired the growth of autonomous racing research (Betz et al., 2022). Researchers are developing algorithms for high-performance race vehicles which aim to operate autonomously at the edge of the vehicle's limits. Competitions in autonomous racing have been held not only in simulation (Hartmann et al., 2021; Babu and Behl, 2020), but also on prototypes ranging from 1:43 scale RC cars (Carrau et al., 2016) to 1/10 scale (O'Kelly et al., 2019), and to full-size Indy racecars (Wischnewski et al., 2022; Jung et al., 2023). To optimize racecars' performance and control for autonomous racing, it is essential to have a model that can precisely predict the vehicle's dynamics. Existing models (Althoff et al., 2017) range from simple point mass vehicle models and single-track models to complex multi-body vehicle models. However, the construction of high-fidelity vehicle dynamics models based on physics is difficult due to the capture of nonlinear behavior of components such as tires, suspension and steering systems. Moreover, obtaining precise parameter values for these components can be expensive and time-consuming. For example, obtaining tire parameters such as tire stiffness, slip angle, slip ratio, etc. involves a combination of performing tire testing experiments on tires test rigs and building mathematical models such as the Pacejka tire model (Pacejka, 2005). Uncertain factors such as road conditions and driver inputs, along with complex subsystems like suspension and steering systems, further complicate the modeling process. Consequentially, researchers have shown interest in using learning-based approaches to address model-output discrepancies (Xing et al., 2020; Hermansdorfer et al., 2020). Specifically, Van Niekerk et al. (2017) proposed Gaussian Processes (GP) models for vehicle dynamics learning,

7th Conference on Robot Learning (CoRL 2023), Atlanta, USA.

which has been further explored by other researchers (Jain et al., 2020; Kabzan et al., 2019; Hewing et al., 2018). However, existing work faces limitations, including scalability issues with GP models, reliance on simplified simulations, arbitrary GP kernel choices or no exploration of kernel learning, and the use of unrealistic race tracks.

This paper proposes DKL-SKIP, a solution to address the mismatch between observations and state predictions obtained with an extended-kinematic single-track model for vehicle dynamics modeling. The main contributions of our paper are stated as follows:

1. We present DKL-SKIP, a new method that integrates deep kernel learning with the scalable SKIP-GP approach for vehicle dynamics modeling. This combination not only overcomes the scalability challenges associated with the traditional GP but also enhances the robustness to noise, and irrelevant features.

2. We evaluated DKL-SKIP on datasets collected from a full-scale autonomous Indy racecar in both real-world and simulating racing environments. We demonstrate our method's capability to accurately capture the non-linear dynamics of a racecar, especially at the limits of its performance envelope.

## 2 Related Work

### 2.1 Gaussian processes for vehicle dynamics modeling

Hewing et al. (2018) pioneered the use of GP models to learn vehicle dynamics in a miniature autonomous racecar, demonstrating their effectiveness in capturing lateral dynamics and developing a model predictive controller. Building on this, Kabzan et al. (2019) applied GP models to learn vehicle dynamics in a driverless electric racecar and developed a contouring model predictive control approach. Additionally, Jain et al. (2020) utilized exact GP models for vehicle dynamics learning in autonomous racing simulations at both 1:43 and 1:10 scales.

However, these methods have the following limitations: (i) The conducted experiments were limited to a small-scale or formula student racecar (Kabzan et al., 2019) unlike our use of a full-sized fully autonomous racecar. (ii) Lack of real-world track layouts, limiting model generalizability. Ning and Behl (2023) have demonstrated that racetrack configurations and choices highly influence the accuracy of the GP-based model. In this paper, we demonstrate that our method works on data obtained from a real racecar running at real racetracks. (iii) The exact GP used in Bayesrace is not scalable to the volume of real-world data samples. In this paper, the training data is 50 times larger, rendering the Bayesrace method intractable. (iv) Previous work lacks of kernel learning exploration, and authors arbitrarily used kernel functions such as RBF or Matérn kernel, while research indicates (Ning and Behl, 2023) that the choice of the kernel can substantially impact GP model performance. Therefore, in this work, we implement a deep neural network (DNN) for kernel learning for the GP models.

### 2.2 Gaussian processes with deep learning

The idea of deep kernel learning, which combines deep learning with Gaussian processes, was introduced by Wilson et al. (2016), who demonstrated the potential of DKL-GP to achieve state-of-the-art results on several benchmarks. Bradshaw et al. (2017) investigated the robustness of DKL-GP models against adversarial examples and transfer learning scenarios. Although the DKL-GP method has been employed for robotics and control tasks (Lee et al., 2022), to the best of our knowledge, this paper is the first work of integrating GPs with deep learning approaches for vehicle dynamics modeling. Specifically, we propose DKL-SKIP to learn the dynamics of a full-sized, real-world racecar by incorporating deep kernel in conjunction with sparse GPs.

| Notation | Vehicle Dynamics | | |
|---|---|---|---|
| | Kinematic | Dynamic | E-Kin |
| $x,y$: Vehicle position | | | |
| $\delta$: Steering angle $[rad]$ | $\dot{x} = v\cos\psi$ | $\dot{x} = v_x\cos\psi - v_y\sin\psi$ | $\dot{x} = v_x\cos\psi - v_y\sin\psi$ |
| $\psi$: Heading angle $[rad]$ | $\dot{y} = v\sin\psi$ | $\dot{y} = v_x\sin\psi + v_y\cos\psi$ | $\dot{y} = v_x\sin\psi + v_y\cos\psi$ |
| $\omega$: Yaw rate $[rad/s]$ | $\dot{\delta} = \Delta\delta$ | $\dot{\delta} = \Delta\delta$ | $\dot{\delta} = \Delta\delta$ |
| $l_f, l_r$: C.O.G. to front/rear axle [m] | $\dot{\psi} = \frac{v}{l_r + l_f}\tan\delta$ | $\dot{\psi} = \omega$ | $\dot{\psi} = \omega$ |
| $v_x$: Longitudinal velocity $[m/s]$ | $\dot{v} = a_x$ | $\dot{v}_x = \frac{1}{m}(F_x - F_{fy}\sin\delta + mv_y\omega)$ | $\dot{v}_x = a_x$ |
| $v_y$: Lateral velocity $[m/s]$ | | $\dot{v}_y = \frac{1}{m}(F_{ry} + F_{fy}\cos\delta - mv_x\omega)$ | $\dot{v}_y = \frac{l_r}{l_r + l_f}(a_x\psi + v_x\omega)$ |
| $a_x$: Longitudinal acceleration $[m/s^2]$ | | $\dot{\omega} = \frac{1}{I_z}(l_f F_{fy}\cos\delta - l_r F_{ry})$ | $\dot{\omega} = \frac{a_x}{l_r + l_f}\sin\delta$ |
| $F_{fy}, F_{ry}$: Tire lateral forces | $F_{fy} = D\sin(C\arctan(B\alpha_f - E(B\alpha_f - \arctan(B\alpha_f))))$ | | |
| $F_x$: Longitudinal force | $F_{ry} = D\sin(C\arctan(B\alpha_r - E(B\alpha_r - \arctan(B\alpha_r))))$ | | |

Table 1: Mathematical descriptions of different vehicle dynamics models. The vehicle dynamics models have three components:(1) the kinematic single-track model, (2) the dynamic single-track model, and (3) the extended-kinematic model used in this paper.

# 3  Vehicle Dynamics Background

Kinematic and dynamic single-track models are widely used based on a good trade-off between simplicity and accuracy. The single-track models simplify the vehicle by lumping the front and rear wheel pairs into one wheel, similar to a bicycle configuration (Althoff et al., 2017). The details of differential equations for the kinematic model are shown in the second column of Table 1. $x$ and $y$ are the coordinates of the vehicle's center of gravity (C.O.G.) with respect to the inertial frame. Velocity at C.O.G. of the vehicle is denoted by $v$. $\psi$ is vehicle inertial heading, and $a_x$ is longitudinal acceleration. $\delta$ is the steering angle that represents the angle between the vehicle heading and the front wheel heading. The model states are $\{x, y, v, \delta, \psi\}$, the inputs to the model are acceleration and steering rate $\{a_x, \Delta\delta\}$, and the parameters of the model are the distance between front axle and vehicle C.O.G., $l_f$, and the distance between rear-axle and vehicle C.O.G., $l_r$.

Although, easier to construct, kinematic models lack accuracy at higher speeds as they neglect tire slip, thereby overlooking crucial effects like understeer and oversteer (Rajamani, 2011). On the other hand, a dynamic single-track model (Fig. 1) takes into account the complex dynamics of a vehicle, including the effects of tire slip and the interaction between tire forces and vehicle motion (Hwan Jeon et al., 2011). It considers additional factors such as lateral acceleration, yaw rate, and tire forces. This model is more accurate and suitable for high-speed maneuvers and scenarios where the vehicle operates closer to its physical limits. The state equations are denoted in the third column of Table 1, where $F_x$ is the longitudinal force, and $F_{fy}$, $F_{ry}$ are the lateral forces at the front axle and rear axle, respectively. In addition to the states in kinematic models, the dynamic mod-

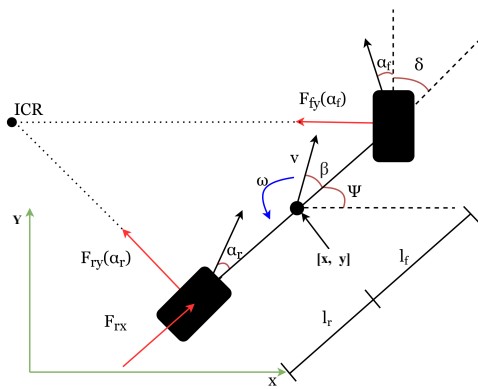

Figure 1: Dynamic single-track vehicle model. Reference point: C.O.G.

els consider lateral speed, $v_y$, as well as the slip angle of the tire, $\alpha_r$ or $\alpha_f$, which is defined as the angle between the heading of the wheel and the velocity vector of the wheel, and used for deriving tire lateral forces $F_{fy}$, $F_{ry}$ using the Pacejka equations (bottom rows in Table 1). However, the model coefficients, $BCDE$, are expensive to obtain and need to be recalibrated every time for every new racetrack, which increases the cost of implementing such a dynamic model.

**Extended kinematic model** In this work, we use an extended kinematic single-track (E-kin) model, as shown in Table 1, which preserves the simplicity of a kinematic model but has identical states to a dynamic model. However, the states of the E-kin model and dynamic model differ in velocities and yaw rate, $\{\dot{v}_x, \dot{v}_y, \dot{\omega}\}$. Specifically, unlike the dynamic models, the E-kin model uses measurable variables, such as $a_x, \psi, \delta$, and two parameters, $l_f, l_r$, to approximate the longitudinal and lateral forces. Therefore, the E-kin model requires less effort in model calibration than a dynamic single-track model. On the other hand, these approximations cause the E-kin model to diverge from the real dynamics. Therefore, when using the E-kin model to estimate the vehicle dynamics, we employ GP models to calculate the discrepancies between the E-kin model and the racecar observations.

## 4 Problem Statement

In this paper, our goal is to construct an extended kinematic single-track model for an autonomous racecar and use GPs to learn the mismatch between the E-kin model and the observed racecar dynamics. In particular, we use racecar state measurements, denoted by set $D := \{d_1, ..., d_n\}$ for $n$ data samples, to address the differences between the E-kin model and the observed racecar dynamics. Every data sample, $d_i$, consists of states, $s_i := \{x_i, y_i, v_{xi}, v_{yi}, \omega_i, \psi_i, \delta_i\}$, and inputs, $u_i := \{a_{xi}, \Delta\delta_i\}$. The E-kin model is represented as $f_{Ekin}$, and the model output at time step $t$ is created using the identical states, and inputs from the racecar measurements noted as $f_{Ekin}(s_t, u_t)$. Therefore, at time $t$, the model residual $\mathbf{r}_t$ can be expressed as in Eq(1).

$$\mathbf{r}_t = d_{t+1} - f_{Ekin}(s_t, u_t) \tag{1}$$

We then use GPs for model residual learning:

$$e(d_t) = \mathcal{GP}(d_t) = \mathbf{r}_t + \nu_t \tag{2}$$

where $e(d_t)$ is the GP function given data sample $d_t$, and $\nu_t \sim \mathcal{N}(0,1)$ is the additive Gaussian noise. By doing so, we can construct a corrected dynamics model, $f_{corr}$, which is the combination of $f_{Ekin}$ and GP, to approximate the racecar's observed dynamics. For instance, at time $t$, the corrected model $f_{corr}(s_t, u_t)$ can be used to approximate racecar's observed dynamics $d_{t+1}$ as:

$$f_{corr}(s_t, u_t) = f_{Ekin}(s_t, u_t) + e(d_t) \approx d_{t+1} \tag{3}$$

As shown in Table 1, it is worth noting that the model differences between the observation and E-kin model outputs only occur in the velocity, $v_x, v_y$, and yaw rate $\omega$ states. This implies that, in this work, $\mathbf{R} = [0, 0, \epsilon_{v_x}, \epsilon_{v_y}, \epsilon_\omega, 0, 0, 0, 0]$, $\mathbf{R}$ is the set of model residuals, $\mathbf{R} := \{\mathbf{r}_1, ...\mathbf{r}_{n-1}\}$ Therefore, we learn three GP models, namely $e_{v_x}, e_{v_y}, e_\omega$, to approximate model residuals in $\epsilon_{v_x}, \epsilon_{v_y}, \epsilon_\omega$. In the next section, we describe the implementation of the DKL-SKIP method to learn these GPs.

## 5 DKL-SKIP

In this section, we describe the DKL-SKIP method, which is a combination of deep kernel learning and a scalable GP, SKIP-GP. This combines the advantages of both DKL and SKIP-GP, and allows us to apply it to capture the non-linear relationship for vehicle dynamics modeling mismatch.

### 5.1 Deep kernel learning

The goal of deep kernel learning is to learn a kernel function for GPs. This is achieved by applying a deep learning neural network for feature extraction before it is fed into the GPs, transforming the inputs into a higher-level representation. In GP, a kernel function is used to measure the similarity between inputs, which influences the predictions of GP models. One of the most commonly used kernel functions in GP is the RBF kernel, denoted in Eq( 4).

$$k(d_i, d_j) = \sigma^2 \exp\left(-\frac{||d_i - d_j||^2}{2l^2}\right), i, j \in \{1, ..., n\} \tag{4}$$

where $d_i, d_j$ are the input data sample, i.e., racecar state measurements in this paper, $\sigma^2$ is the variance, $||d_i - d_j||$ is the Euclidean distance between inputs, $l$ is the length-scale. Further details of GP can be referred to Rasmussen (2003). In this paper, we transform a kernel $k(d_i, d_j; \theta)$ to a deep kernel function, as shown in Eq( 5).

$$k(d_i, d_j; \theta) \rightarrow k(g(d_i, w), g(d_j, w) | \theta, w) \tag{5}$$

where $g(d, w)$ is a deep learning model, such as a deep neural network. $\theta$ is the kernel hyperparameters, e.g., the length-scale ($l$) of RBF kernel, and the $w$ is the DNN weight. The deep kernel hyperparameters, denoted as $\gamma = \{\theta, w\}$, can be learned jointly by maximizing the log marginal likelihood. Moreover, DKL acts as a feature extractor, which transfers each input sample ($d$) into a lower-dimensional feature expression ($\tilde{d}$). Hence, the transformation of a kernel into a deep kernel can then be expressed as in Eq(6). This enables DKL to capture the most representative information from the input data as well as reduce the data dimensionality.

$$k(d_i, d_j; \theta) \rightarrow k(\tilde{d}_i, \tilde{d}_j; \gamma) \tag{6}$$

## 5.2 SKIP-GP

The kernel learning process of GPs involves using the Cholesky decomposition of the kernel matrix, $K_{DD}$, which is a matrix composed of kernel functions of $n$ data points $\{d_1, ..., d_n\}$. In this paper, we use DKL to transform input data into lower-dimensional features ($\tilde{d}$) leading to a deep kernel matrix, $K_{\tilde{D}\tilde{D}}^{deep}$ which improves the scalability of GPs. However, computing this decomposition requires matrix-vector multiplies (MVMs), which still leads to $\mathcal{O}(n^3)$ time and $\mathcal{O}(n^2)$ space complexity (Rasmussen, 2003). Therefore, we use SKIP-GP, proposed by Gardner et al. (2018), to reduce the computing complexity. There are two key components in SKIP-GP: (i) Structured kernel interpolation (SKI) (Wilson and Nickisch, 2015); and (ii) Product kernel structure. First, given a set of $m$ inducing points, $U = \{\tilde{d}_1, ..., \tilde{d}_m\}$, where $m \ll n$, the SKI computes kernel functions between inducing points and then approximates the true kernel functions:

$$k(\tilde{d}_i, \tilde{d}_j) \approx \mathbf{w}_{\tilde{d}_i} K_{UU} \mathbf{w}_{\tilde{d}_j}^T, i, j \in \{1, ..., m\} \tag{7a}$$

$$K_{\tilde{D}\tilde{D}}^{deep} \approx \mathbf{W}_{\tilde{D}} K_{UU} \mathbf{W}_{\tilde{D}}^T \tag{7b}$$

where $\mathbf{W}_{\tilde{D}}$ is the sparse matrix composed of sparse vector ($\mathbf{w}_{\tilde{d}}$) that contains approximation weights. And thus, Eq(7b) denotes applying the approximation for all data. This method allows SKI to achieve linear time and storage complexity. However, it leads SKI to an exponential time complexity in the dimensionality of the inputs. The SKIP-GP uses product kernel structure to address the curse of dimensionality in SKI: given the data with $\mathbf{p}$ dimensions, the kernel matrix of the product kernel structure can be expressed as $K_{\tilde{D}\tilde{D}}^{deep} = K_{\tilde{D}\tilde{D}}^{(1)} \times ... \times K_{\tilde{D}\tilde{D}}^{(\mathbf{p})}$ where $\times$ is element-wise multiplication. And each component, $K_{\tilde{D}\tilde{D}}^{(i)}$, is approximated using SKI: $K_{\tilde{D}\tilde{D}}^{(i)} = \mathbf{W}_{\tilde{D}}^{(i)} K_{UU} \mathbf{W}_{\tilde{D}}^{(i)T}$. This ensures SKIP-GP achieves linear time complexity even with high-dimensional inputs.

**Combining DKL with SKIP-GP** by using inducing points to approximate the deep kernel matrix, $K_{\tilde{D}\tilde{D}}^{deep}$, the resulting kernel is denoted as $K_{UU}^{deep}$. Then we can use this inducing kernel matrix to approximate the deep kernel matrix: $K_{\tilde{D}\tilde{D}}^{deep} \approx \mathbf{W}_{\tilde{D}} K_{UU}^{deep} \mathbf{W}_{\tilde{D}}^T$. Therefore, the DKL-SKIP achieves linear time complexity while maintaining robustness and expressiveness.

## 6 Experiment Setup

We validate our method using both real-world data and data obtained from AutoVerse, a high-fidelity racing simulator. All experiments were conducted on a Linux-based system equipped with sixteen 3.8 GHz CPU cores, 16 GB of RAM, and a single NVIDIA GeForce RTX 3080 GPU.

### 6.1 Real-world autonomous racing setup

The real data is obtained from a full-scale, fully autonomous racecar, shown in Figure 2 Ⓐ . These racecars are engineered for the high-speed, autonomous racing competition in the Indy Autonomous

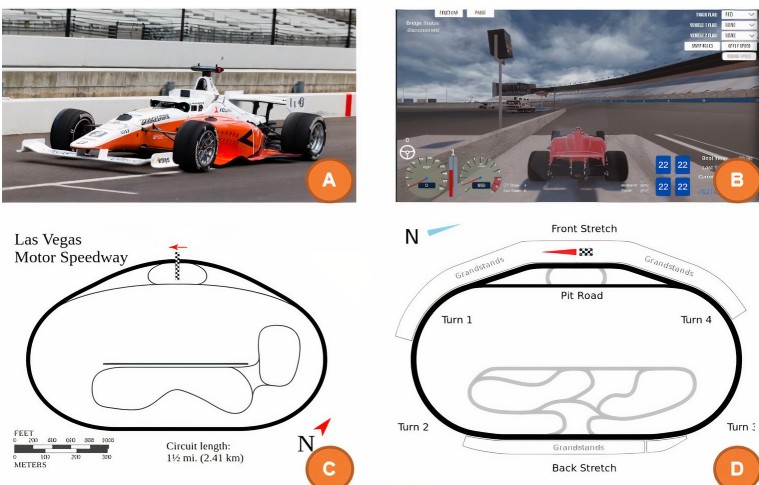

Figure 2: Experiment setup: (A) IAC full-sized, fully autonomous racecar, on (C) Las Vegas Motor Speedway race track; (B) IAC racecar in AutoVerse simulator, with (D) race track in Texas Motor Speedway .

Challenge (IAC). These vehicles have been customized with sensors such as LiDAR, GNSS, cameras, and radar, computing hardware, and other autonomy-enabling components. The data was collected during solo runs at speeds $80\,\mathrm{mph}$ at the Las Vegas Motor Speedway, Figure 2 (C). During these runs, state measurements were obtained by the Extended Kalman filter algorithm and logged as ROS2 bag files. These bag files were processed to construct the data set, $D_{real}$, consisting of $\{v_x, v_y, \psi, \delta, \omega, a_x, \Delta\delta, u_T, u_B\}$, which are longitudinal and lateral velocities, vehicle heading, yaw rate, steering angle, longitudinal acceleration, steering velocity, throttle command, and braking pressure. This dataset is recorded at $25\,\mathrm{Hz}$, divided into a training set encompassing 15,789 samples (631 seconds) and a testing set, unseen to the model, comprising 6,432 data samples (257 seconds).

## 6.2 Simulated autonomous racing setup

For more repeatable analysis, we also conduct experiments using AutoVerse, designed to replicate the racing environment for data collection (Autonoma, 2023). As shown in Figure 2 (B), AutoVerse can simulate vehicle simulator for the IAC racecars. We set up a single racecar scenario on the Texas Motor Speedway racetrack, shown in Figure 2 (D), where the car utilizes a pure pursuit algorithm and runs at higher speeds, $130\,\mathrm{mph}$, than those recorded during real-world data collection. This approach aims to validate our method's performance in learning vehicle dynamics at the limits of the vehicle's performance. The data set, $D_{sim}$, recorded at $25\,\mathrm{Hz}$ is composed of training set consisting of 11,737 data samples (469 seconds), the testing set has 7,153 data samples (286 seconds).

## 6.3 DKL-SKIP setup

Here, we describe the setup of deep neural network feature extractor used to define the deep kernel of the GP model. In this work, we used a fully connected DNN with a $[9-800-300-50-4]$ architecture, where the numbers indicate the number of neurons in each layer. Specifically, the first layer has 9 inputs corresponding to 9 features in the dataset, which are $\{v_x, v_y, \psi, \delta, \omega, a_x, \Delta\delta, u_T, u_B\}$. The final layer has 4 outputs, which is determined based on the experiments and means that the DNN maps to 4 final features. These output features are then passed to the SKIP-GP. ReLU activation layers and dropout layers with $20\%$ rate have been integrated into the DNN. Moreover, we consider the scalable deep kernel learning with the RBF base kernel, as shown in Eq(4), which is a popular choice of base kernel. During the training process, we run 60 epochs of training using the Adam optimizer, and the learning rate is $0.02$. Besides, both the hyperparameters of the deep kernel, $\theta$, and the parameters of the DNN, $w$, are learned using Type-II maximum likelihood estimation.

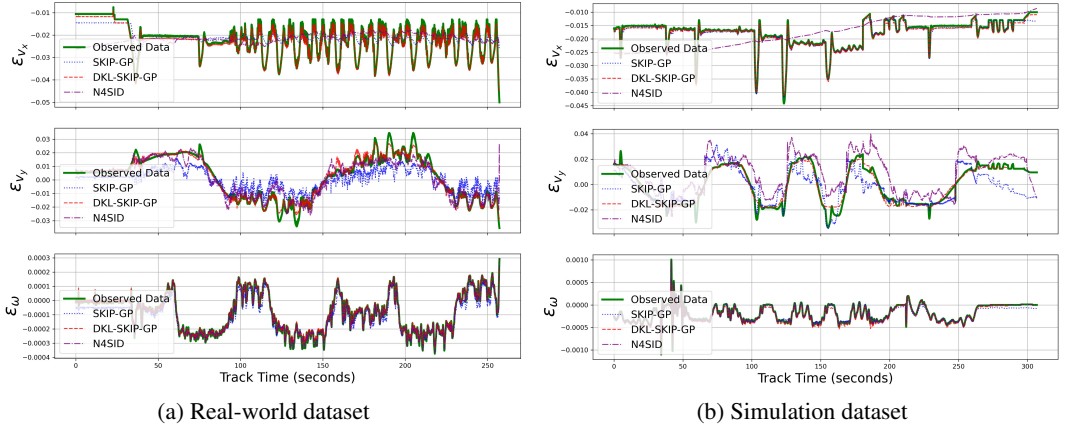

(a) Real-world dataset          (b) Simulation dataset

Figure 3: Comparison of prediction performance among various models using mean prediction values: (1) The SKIP-GP model is represented by blue dotted lines; (2) Red dashed lines indicate the DKL-SKIP model; and (3) The N4SID model is shown by purple dashed lines.

## 6.4 Data preparation

To address the mismatch between the real racecar observations and the extended kinematic model data, we use the collected data $D_{exp}$, comprising of both the real-world data and the simulator data, $D_{exp} \equiv D_{real} \cup D_{sim}$. Given the parameters $(l_f, l_r)$ of the E-kin model $f_{Ekin}$ are known, we can construct a dataset, $D_{Ekin}$, to capture the model outputs when excited with the same inputs and initial conditions, $D_{Ekin} = \{f_{Ekin}(s_i, u_i)\}, \forall i \in \{1, \dots, n\}$, where $s_i, u_i$ represent data sample of state measurements and control input of $D_{exp}$. For GPs training and testing, the input states consist of $\{v_x, v_y, \psi, \delta, \omega, a_x, \Delta\delta, u_T, u_B\}$, and the outputs are $\epsilon_{v_x}, \epsilon_{v_y}, \epsilon_\omega$, respectively.

# 7 Results

In this section, we evaluate DKL-SKIP, SKIP-GP, and a system identification technique, N4SID, proposed by Van Overschee and De Moor (1994). Our comparison focuses on their capabilities in predicting the mismatch between the extended kinematic single-track model and observation.

## 7.1 Real-world results

Figure 3a shows the comparison between DKL-SKIP, SKIP-GP and N4SID for each error term $(\epsilon_{v_x}, \epsilon_{v_y}, \epsilon_\omega)$, and we can see that DKL-SKIP is able to accurately predict the mismatch between observed states and E-kin model output. In addition, we compare these models on mean absolute error, root mean square error, normalized root mean square error, and the coefficient of determination $(R^2)$, left table of Table 2. When examining the $R^2$, DKL-SKIP outperforms N4SID and SKIP-GP methods by $99\%, 62\%$ for error $\epsilon_{v_x}$ and $24\%, 32\%$ for error $\epsilon_{v_y}$, respectively. N4SID slightly outperforms DKL-SKIP in the $\epsilon_\omega$ error term by a margin of $0.16\%$ in terms of the coefficient of determination. However, it does poorly at predicting the other two error terms. Overall, the results on real data show DKL-SKIP superior predictive capabilities across the three error terms.

## 7.2 Simulation results

The comparison between the models on simulated data is shown in Figure 3b and in the right side of Table 2. Once again, DKL-SKIP model consistently performs well across all error terms. This indicates the effectiveness of DKL-SKIP and its ability to predict the non-linear error terms under high-speed situations. In terms of $R^2$, DKL-SKIP outperforms N4SID and SKIP-GP methods by $99\%, 2\%$ for error $\epsilon_{v_x}$ and $63\%, 34\%$ for error $\epsilon_{v_y}$, respectively. The SKIP-GP is able to make accurate predictions on error terms $\epsilon_{v_x}$ and $\epsilon_\omega$, but it is not as accurate in predicting $\epsilon_{v_y}$. Although

**Errors in long. velocity, $\epsilon_{v_x}$ (real-world)**

| Method | MAE | RMSE | NRMSE | $R^2$ |
|---|---|---|---|---|
| N4SID | 0.00527 | 0.00680 | 0.15956 | 0.0020 |
| SKIP-GP | 0.00405 | 0.00538 | 0.12662 | 0.3720 |
| DKL-SKIP | 0.00103 | 0.00111 | 0.02607 | 0.9733 |

**Errors in long. velocity, $\epsilon_{v_x}$ (Simulation)**

| Method | MAE | RMSE | NRMSE | $R^2$ |
|---|---|---|---|---|
| N4SID | 0.00754 | 0.00882 | 0.26283 | -2.463 |
| SKIP-GP | 0.00094 | 0.00094 | 0.02801 | 0.9606 |
| DKL-SKIP | 0.00057 | 0.00064 | 0.01874 | 0.9823 |

**Errors in lateral velocity, $\epsilon_{v_y}$ (real-world)**

| Method | MAE | RMSE | NRMSE | $R^2$ |
|---|---|---|---|---|
| N4SID | 0.00609 | 0.00787 | 0.11236 | 0.7347 |
| SKIP-GP | 0.00730 | 0.00891 | 0.12733 | 0.6590 |
| DKL-SKIP | 0.00172 | 0.00246 | 0.03516 | 0.9739 |

**Errors in lateral velocity, $\epsilon_{v_y}$ (Simulation)**

| Method | MAE | RMSE | NRMSE | $R^2$ |
|---|---|---|---|---|
| N4SID | 0.00985 | 0.01215 | 0.20140 | 0.3230 |
| SKIP-GP | 0.00816 | 0.00926 | 0.15363 | 0.6141 |
| DKL-SKIP | 0.00216 | 0.00301 | 0.04998 | 0.9592 |

**Errors in yaw rate, $\epsilon_{\omega}$ (real-world)**

| Method | MAE | RMSE | NRMSE | $R^2$ |
|---|---|---|---|---|
| N4SID | 4.96e-6 | 6.78e-6 | 0.010001 | 0.9985 |
| SKIP-GP | 3.83e-5 | 4.51e-5 | 0.06757 | 0.8837 |
| DKL-SKIP | 5.77e-6 | 7.26e-6 | 0.01088 | 0.9969 |

**Errors in yaw rate, $\epsilon_{\omega}$ (Simulation)**

| Method | MAE | RMSE | NRMSE | $R^2$ |
|---|---|---|---|---|
| N4SID | 2.71e-06 | 9.47e-06 | 0.00448 | 0.9971 |
| SKIP-GP | 3.91e-05 | 5.18e-05 | 0.02456 | 0.9133 |
| DKL-SKIP | 1.95e-05 | 2.9e-05 | 0.01374 | 0.9829 |

Table 2: Evaluation of N4SID, SKIP-GP, DKL-SKIP (our method), in real-world setup (left table) and simulation setup (right table): The tables show these models' performance evaluated in four metrics, such as Mean Absolute Error (MAE), Root Mean Square Error (RMSE), Normalized Root Mean Square Error (NRMSE), and Coefficient of Determination ($R^2$).

N4SID marginally performs better on $e_{\omega}$, by $1\%$, it again struggles to accurately predict the other error terms, particularly $\epsilon_{v_x}$. DKL-SKIP's ability to provide accurate predictions under conditions when the vehicle operates at high speeds further demonstrates its robustness and effectiveness in learning the mismatch between Ekin model and observed dynamics for autonomous racing.

## 8 Limitations

The DKL-SKIP model has shown promise in learning vehicle dynamics in the context of autonomous racing, but a few limitations are noteworthy. First, DKL-SKIP still faces computational challenges, mainly impacted by the deep learning component, which is computationally expensive, especially for large datasets. In addition, the SKIP-GP has a $\mathcal{O}(dkn)$ space complexity which will limit GPU performance when the number of copies of dataset, $k$, increases. These factors limit the model's performance in real time. In this paper, without any code optimization, DKL-SKIP has an inference time ranging from $20\,\text{to}\,40\,\text{ms}$. Since it has not been verified in closed-loop evaluations, and thus, the computing time may limit its suitability for real-time prediction. This will be addressed in future work as we implement DKL-SKIP-based model predictive control for autonomous racing. Second, as a supervised learning method, the predictive accuracy of DKL-SKIP depends on the distribution of the training data. The model's performance will be impaired by unseen input data samples, such as abrupt changes in vehicle speed or direction. Finally, the states $v_x, v_y, \omega$ in the E-kin model are simply approximations. We have not explored different variable settings in their differential equations, which could impact DKL-SKIP prediction ability.

## 9 Conclusion and Discussion

This paper introduces DKL-SKIP, a scalable GP model combined with the deep kernel learning for learning the mismatch between an Ekin model and observed dynamics for autonomous racing. We conduct both real-world and simulation experiments to compare DKL-SKIP against SKIP-GP, and N4SID. In real-world evaluation, in terms of $R^2$, DKL-SKIP outperforms competitors by $99\%, 62\%$ for $\epsilon_{v_x}$ and $24\%, 32\%$ for $\epsilon_{v_y}$, respectively. DKL-SKIP also outperforms other models on simulated data by $99\%, 2\%$ for $\epsilon_{v_x}$ and $63\%, 34\%$ for $\epsilon_{v_y}$, respectively. Besides, DKL-SKIP performs well on predicting $\epsilon_{\omega}$, specifically, in terms of $R^2$, archives 0.9969 in real-world data, and 0.9829 in simulated data. Our future work involves implementing model predictive control and evaluation in closed-loop experiments as well as extending this methodology to multi-agent autonomous racing.

**Acknowledgments**

This material is based upon work supported by the National Science Foundation under Grant No. 2046582.

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

In this supplementary material, we present a generalized overview of the N4SID system identification technique. In addition, we include the details of the ablation studies that were undertaken in the development of DKL-SKIP.

## A   N4SID System Identification Algorithm

In this paper, we implement an alternative technique known as N4SID, a data-driven approach designed for system identification (Van Overschee and De Moor, 1994). This method utilizes a subspace-based strategy, which segregates the data into deterministic and stochastic elements by projecting them onto distinct orthogonal subspaces. The algorithm calculates the system's state sequence, state-transition matrid, input matrid, and output matrix from these subspaces, offering a concise and effective depiction of the inherent dynamical system. Essentially, N4SID is capable of directly approximating system behavior using input-output data in a state-space format. A description of the algorithm is shown below:

---
**Algorithm 1** N4SID Algorithm

---
1:  **procedure** N4SID
2:      Normalize input-output data.
3:      Estimate a covariance matrix by applying QR decomposition.
4:      Compute a singular value decomposition (SVD) of the covariance matrix.
5:      Divide the SVD output into observable and unobservable subspaces.
6:      From the observable subspace, compute the system matrices $A$, $B$, $C$, and $D$.
7:      Perform the balancing transformation and reduction of the state-space model.
8:      Return the state-space model.
9:  **end procedure**

---

In this paper, considering the collected input-output data set $u(k), y(k)_{k=1}^{N}$, with $u(k) \in \mathbb{R}^m$ representing the input data sample, $\{v_{xk}, v_{yk}, \psi_k, \delta_k, \omega_k, a_{xk}, \Delta\delta_k, u_{Tk}, u_{Bk}\}$, and $y(k) \in \mathbb{R}^p$ denoting the output vector, $\epsilon_{v_{xk}}, \epsilon_{v_{yk}}, \epsilon_{\omega_k}$, at time $k$, N4SID can effectively approximate the system's dynamics using a state-space representation, as demonstrated in 8.

$$\begin{cases} x(t+1) = Ax(k) + Bu(k) \\ y(t) = Cx(k) + Du(k) \end{cases} \tag{8}$$

## B   DKL-SKIP setup

### B.1   Architecture of neural network

The architecture of our DNN was inspired by the work on Deep Kernel Learning by Wilson et al. (2016). While we adopted their layered approach, we made modifications to make it suitable for our tasks.

One key adaptation is the configuration of the first layer, which comprises nine neurons, mirroring the nine input features in our dataset. The output layer dimensionality is decided through experimentation. Through our tests, we identified an optimal configuration for the output layer that balances maintaining prediction accuracy and simplifying the computational task. Specifically, we reduced the output dimensionality to four neurons. This design choice aids the SKIP-GP model in producing more accurate predictions by reducing the complexity it needs to handle. This is also the reason our DKL-SKIP performs better than SKIP-GP in terms of prediction accuracy.

The number of hidden layers (network capacity) was decided based on the experiment. Below, we compare the R2 score as model performance across different architectures, and the results are shown in Table 3. As shown in the results, it is clear to see that as the capacity of the DNN is augmented, there's a typical trend of enhancement in our model's performance. This improvement is attributed

| Architecture | R2 scores ($v_x$) | R2 scores ($v_y$) | R2 scores ($\omega$) |
|:---:|:---:|:---:|:---:|
| $9 \rightarrow 50 \rightarrow 4$ | 0.96095 | 0.95242 | 0.89471 |
| $9 \rightarrow 300 \rightarrow 50 \rightarrow 4$ | 0.97435 | 0.96716 | 0.99369 |
| $9 \rightarrow 800 \rightarrow 300 \rightarrow 50 \rightarrow 4$ | 0.97489 | 0.96347 | 0.99522 |
| $9 \rightarrow 1200 \rightarrow 800 \rightarrow 300 \rightarrow 50 \rightarrow 4$ | 0.95730 | 0.93647 | 0.91790 |

Table 3: R2 scores for different architectures. As DNN capacity increases, performance generally improves, but it can decline if capacity becomes excessive.

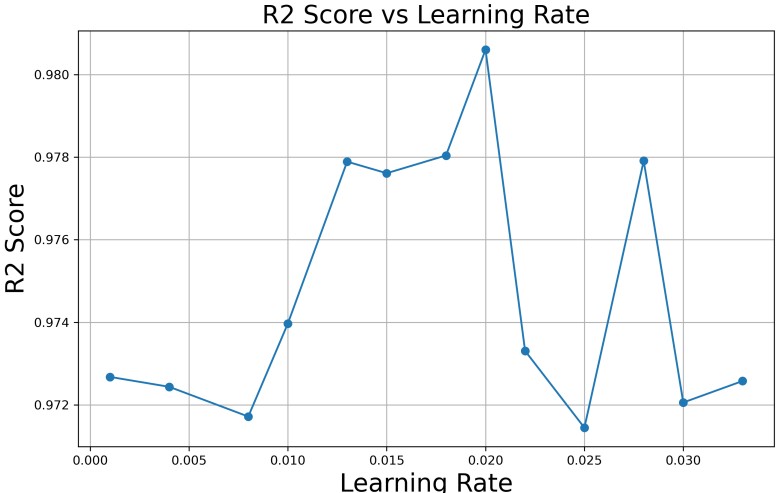

Figure 4: Performance comparison across different learning rates (0.001 to 0.035) with the optimal R2 score of 0.9806 achieved at a rate of 0.02.

to the increased ability of the DKL to capture complex patterns and relationships in the dynamics data. However, it's essential to strike a balance. If the DNN's capacity becomes overly large, it may start to overfit the training data, which eventually degrades the overall performance of our model.

In addition, we also carried out an ablation study to determine the optimal learning rate for tuning the DNN and the Gaussian Process (GP) kernel function. The results are shown in Figure 4. In our study, we executed a range of experiments, testing learning rates between 0.001 and 0.035. We used the R2 score as our benchmark for model performance. Our findings revealed that a learning rate of 0.02 yielded the highest performance, achieving an R2 score of 0.9806. Moreover, this rate ensured consistent stability during training.

## B.2 Input data features

In this paper, the DKL-SKIP model was not exposed to the testing data during the training process. The training and testing data were collected during distinct runs on the racetrack. The variations in throttle/braking and steering between these datasets resulted in differing vehicle dynamics. To offer a clearer understanding, we've depicted each input data set in Figure 5.

The input features of our data sets consisting of $\{v_x, v_y, \psi, \delta, \omega, a_x, \Delta\delta, u_T, u_B\}$, representing longitudinal and lateral velocities, vehicle heading, yaw rate, steering angle, longitudinal acceleration, steering velocity, throttle command, and braking pressure, respectively.

These features serve as the initial input for the DKL feature extractor. Within the DKL framework, a series of processes occur: First, the dimensionality of this input data is significantly reduced to simplify the data structure. This condensed data is then transformed, mapping it to a more abstract

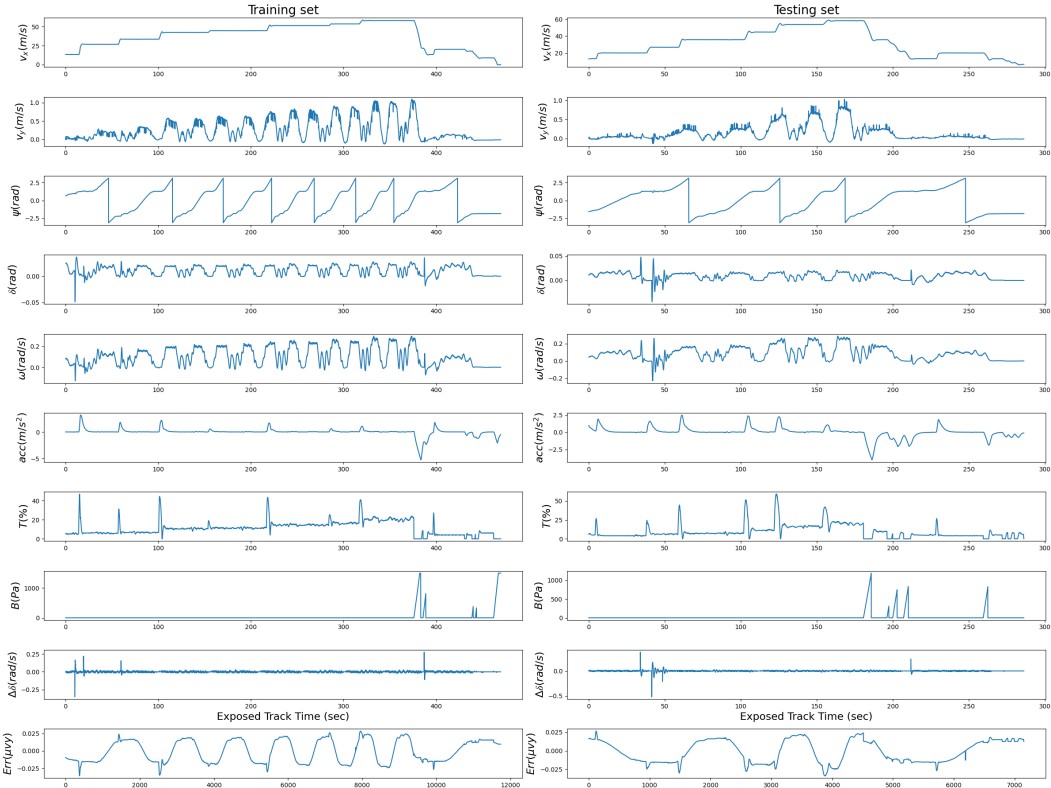

Figure 5: Distinct racetrack runs for training and testing data collection, highlighting differences in vehicle dynamics due to variations in throttle/braking and steering.

feature space. Out of this transformed data, DKL identifies and selects the top four most significant and representative features. These features are then forwarded as the input for the GP model, which is responsible for making the final target predictions.

