# OpenReview forum: "Scalable Deep Kernel Gaussian Process for Vehicle Dynamics in Autonomous Racing"
_robot-learning.org/CoRL/2023/Conference — CoRL 2023 Poster_

### Official Review · Reviewer_Fjkf · 2023-07-16

**Confidence:** 5
**Originality:** Fair
**Technical Quality:** Fair
**Clarity Of Presentation:** Good
**Impact:** 2

**Recommendation:**

Strong Reject: I recommend rejecting the paper and will argue for my recommendation even if other reviewers hold a different opinion.

**Review:**

Strengths: The experiment is based on data collected on a real-world track with a full-scale autonomous racecar, which could potentially provide exceptionally valuable insights into advancing real-world autonomous racing algorithms.

Weakness:
1. This work has very limited novelty and contribution in methodology. The proposed model is a straightforward combination of existing deep kernel learning with sparse GP, and GP for racing dynamics model learning has also been widely studied in the literature. The uniqueness of this work mainly lies in its experimental platform, built on a real-world track with a full-scale autonomous racecar.
2. Unfortunately, the current experimental study is very limited and insufficient to validate the effectiveness of the proposed method. The proposed DKL-SKIP is evaluated on offline collected data. The data distribution highly depends on the controller and the reference path/speed adopted to drive the racecar. More importantly, its distribution could be significantly different from the trajectory distribution under closed-loop control (i.e., controlling the racecar using the learned GP model). Thus, the evaluation results on offline data are insufficient to validate the robustness of the proposed method.

**Quality Of The Limitations Section:**

Limitations are addressed clearly

**Questions For Rebuttal:**

It seems that the GP model is only used to estimate the mean of the model residual and the randomness is injected with a unit Gaussian noise. Then why can't the authors simply replace the GP model with a neural network (e.g., MLP or RNN), especially considering that the proposed method already incorporates a neural network for kernel learning?

**Robotics Focus:**

Highly relevant to robotics but no hardware experiments

**Summary Of Paper:**

This paper combined deep kernel learning with Gaussian Process to develop a learning-based residual dynamics model for autonomous racing. The proposed method is evaluated on offline racing data collected on a real-world race track with a full-scale racecar.

**Summary Of Recommendation:**

Given the limited novelty and insufficient experimental study, I recommend rejecting this paper in its current form. It could potentially be turned into a valuable contribution to the community, if the authors could validate the proposed method by combining it with a model-based controller and evaluating it in a closed-loop fashion on the real-world racing platform.

---

### Official Review · Reviewer_nK5o · 2023-07-19

**Confidence:** 3
**Originality:** Good
**Technical Quality:** Good
**Clarity Of Presentation:** Very Good
**Impact:** 3

**Recommendation:**

Weak Accept: I recommend accepting the paper, but will not argue for my recommendation if the majority of other reviewers have a different opinion.

**Review:**

The idea proposed by this work is generally convincing. The authors try to solve a problem highly relevant to mobile robots and autonomous driving. Though the technical novelty is limited, i.e. combining two existing approaches and there are not much texts on describing the technical difficulties, I think the straightforward combination is valuable in this application scenario. Because it addresses the limitations in previous works, i.e. a learnable kernel matrix and the scalability issue of GPs. More importantly, the validation of the proposed idea has been performed on real data of a full-size racecar, more realistic than test data of only small scale racecar in other related work. Besides, there is another data set in simulation used in the experiements. On the other hand, the presentation is generally clear and concise, I don't have much difficulty on understanding the key ideas. However, there are some suggestions that might be helpful on further improving the manuscript:

### Writing
- Minor: "Gaussian Process" -> "Gaussian Processes";

- The DKL and SKIP-GP might be better to be put into a (sub)section called background or preliminaries. Then the authors can elaborate the proposed combination in a more detailed and formal way followed by the following optimization process. A discussion on the complexity comparison would be more informative for the readers that are interested in this method.

### Methods
- Since there is missing description on SKIP-GP (see above), why does SKIP-GP lead to an exponential time complexity in the dimensionality of the inputs? This should be the another motivation for using DKL on top of SKIP-GP. But from the texts, it is not straightforward to understand.

- Why are the letters for inducing points the same as those for features from DKL? They are both $\tilde{d}_i$.

- As mentioned in the limitation, for real world deployment, out-of-distribution data might occur and cause unexpected behaviors of the model. Since GPs are also good at providing reliable uncertainty estimates. To evaluate the ability of detecting these data can offer more safety guarantee for the autonomous racecar. Have the authors consider this aspect.


**Quality Of The Limitations Section:**

Limitations are addressed clearly

**Questions For Rebuttal:**

I would suggest the authors to refine the writing accordingly and address the concerns in the review above.

**Robotics Focus:**

Highly relevant to robotics but no hardware experiments

**Summary Of Paper:**

The authors proposed to model the dynamics of an autonomous racecar by learning the residuals between an extended kinematic model and the real observations with a Gaussian Process (GP). To overcome the limitations of previous GP-based approaches, this work combines two existing techniques for more flexible and scalable GPs modeling, that are deep kernel learning and one variant of scalable GPs called SKIP-GP. Besides, in conrast to previous work, the proposed method is evaluated on the real data of a full-size racecar and in a high-fidelity simulator. Benefits have been shown against the standard GP and a traditional method via empirical experiments.

**Summary Of Recommendation:**

The idea including the method and its evaluation is sound and of value to the community of GP-based dynamics modeling. Though the technical novelty is limited, I would vote for a weak accept for this work.

---

### Official Review · Reviewer_AEJt · 2023-07-19

**Confidence:** 5
**Originality:** Good
**Technical Quality:** Very Good
**Clarity Of Presentation:** Very Good
**Impact:** 3

**Recommendation:**

Weak Accept: I recommend accepting the paper, but will not argue for my recommendation if the majority of other reviewers have a different opinion.

**Review:**

Quality:
- The paper provides a thorough state of the art and is covering many interesting and important papers in the field of autonomous racing and vehicle dynamics modeling. The authors show clearly which papers adress which method so far e.g. the Gaussian Process papers.
- Overall, the paper is of good merit but lacks some information, structural refinements, and some additional explanations in the method section. The method section is quite short, the additional explanation of the vehicle dynamics backgound (section 3) and Table 1 is a litte bit redundant

Clarity:
-The paper's content is presented in a clear and concise manner, making it easy for readers to comprehend the research objectives, methods, and findings. The language used is precise, avoiding unnecessary jargon, and effectively conveying complex ideas without sacrificing understanding. Moreover, the logical flow of information is well-maintained, with each section naturally leading to the next, enhancing the overall coherence of the paper

Originality:
- The paper is addressing the problem of vehicle dynamics estimation, which is a very well know problem in the ground robotics community and has been done many times. The news value and originality value of the paper lies in the application of the DKL-SKIP method, which is a combination of deep kernel learning and a scalable GP, SKIP-GP. From my view the paper offers therefore a good originality since this approach combines the advantages of both DKL and SKIP-GP, and allows to apply it to capture the non-linear relationship for vehicle dynamics modelling mismatch. If this approach is done right the nonlinearities of spring/damper and tyres can be capture more precisely.
- By displaying the current SoTA in the field, the authors have introduced innovative approaches and setting their work apart from previous research clearly.
- Unfortunately, right now the authors are missing to display their main conributions in order to display the reader more clearly what was done in the paper. I recommend entering the main contributions.

Significance:
- While acknowledging the paper's merits in presenting a nice approach, it is essential to recognize that its significance may be relatively moderate when compared to some existing well-established works in the field. Especially classic modelling of the vehicle dynamics is still state of the art. The parameter identification is here the crucical part which takes time. If the authors can address a litte bit more the real-world value e.g. saving time and still being precise enough. It would be helpful.
- While the paper contributes valuable insights and offers an innovative perspective, it might not drastically revolutionize the current state of research. The approach may resonate with specific audiences in the field of autonomous driving and mobile ground robots or research contexts, making it relevant in those particular domains only.

Strength:
- The approach of Deep Kernel Learning was firstly used in this paper to estimate the vehicle dynamics of a mobile ground vehicle with ackerman steering (car)
- The paper is well written and structure and has a clarity in the reading
- Comprehensive Literature Review: The paper showcases a comprehensive and up-to-date literature review that demonstrates the authors' in-depth understanding of the subject.
- The paper is using real world data from a real vehicle in order to estimate the vehicle dynamics. Many authors are using simulation environments only but here real data is used

Weaknesses:
- The papers method is not explained thoroughly e.g. training setup and DNN setup. The DNN architecture is describe in one sentence which does not explained thoroughly how the method actually works. It is recommended to enter a clear picture of the process of using real-world data, training the DKL, and applying the DKL
-You are missing your discussion section completely. From a scientific paper point of view, you need to enter a critical and reflective discussion. Explain to the reader what is good in your approach and what is bad. Be honest here about what needs to be done in addition to enhancing your system further. In addition, you can draw here conclusions from other authors.
- The result section is quite poor only display three values of the estimation (velocity an yawrate). It makes sense to display the other values too and highlight the best values in Table 2

**Quality Of The Limitations Section:**

Limitations are addressed clearly

**Questions For Rebuttal:**

The experimental setup section is a litte bit poor. I understand that the Indy Autonomous Challenge car is driving on an oval only, but the vehicle dynamics are not explored here thoroughly since the car is driving on similar speeds and jus steering to the left side. The authors need to integrate an experimental setup that shows more dynamic behavior e.g. left/right steering, cornering, braking and steering -> Only this captures the nonlinear dynamics more thoroughly and it is questioned why a simple track like the Las Vegas Motor Speedway is used.

**Robotics Focus:**

Sufficient demonstration on hardware

**Summary Of Paper:**

The authors paper is located in the field of mobile ground robotics, especially autonomous cars. The special focus of the authors work is on autonomous race cars, cars that drive on high speed and high acceleration in dynamic environment completely autonomously. A big importance in this field is modeling and determining the correct vehicle dynamics - especially from a nonlinear point of view e.g. spring/damper dynamics and tyre dynamics. The authors propose an approach called "Deep Learning kernel" in order to model/detect the vehicle dynamics correctly without modeling them with explicit ODEs. The authors use a real-world race car (Indy Autonomous Challenge) which provides a rich dataset and apply their approach to the this data. Ultimately, the approach is applied to a simulation environment an compared to other approach e.g. Gaussian process for Vehicle dynamics estimation.

**Summary Of Recommendation:**

The paper presents the application of a Deep Kernel Learning approach in order to estimate the vehicle dynamic behavior of an autonomous racecar. The authors used real-world data from a racecar to train the DNN and then apply it on the real car and in simulation. The authors demonstrate that their approach can capture the nonlinear behavior of the vehicle much better than other approaches. Unfortunately, the authors miss to have some special evaluation scenarios that clearly demonstrate that their approach can be generalized through many various scenarios. Nevertheless, a solid paper that is addressing questions in the vehicle dynamics and autonomous car community.

---

### Official Review · Reviewer_c6Hr · 2023-07-23

**Confidence:** 3
**Originality:** Fair
**Technical Quality:** Fair
**Clarity Of Presentation:** Fair
**Impact:** 3

**Recommendation:**

Weak Accept: I recommend accepting the paper, but will not argue for my recommendation if the majority of other reviewers have a different opinion.

**Review:**

Strengths
============
- The paper shows experiments on both simulated and real-world data.
- The related work section cites relevant work on Gaussian Process-based residual dynamics identification for autonomous race cars.

Constructive Criticism
==================
- The abstract of the paper introduces the abbreviation GP as Gaussian Processes, but does not provide the context on DKL, SKIP, or N4SID. The readability of the paper could be greatly improved if abbreviations were introduced before they are used.

- The limitation reported in L64 is not an algorithmic limitation. Just because the method presented in this paper has been deployed on a full-scale car does not make the approach better. If there are objective and measurable effects that can only be observed on the full-scale race car, then please list them instead of making such a sweeping statement.

- The related work mentions that previous approaches that identified residual dynamics models from vehicle data used simple GPs with RBF or Matern Kernels, and argues that this choice of kernel functions is suboptimal. It would be interesting to compare with these approaches and report their performance for reference as well. Note that these approaches did not use SKIP-GP.

- It would be interesting to understand if the superior performance of DKL-SKIP is across both the training and testing set or only on the testing set. I.e. is the model generalizing better or does it in general have better approximation capacity? The paper does not clearly state if the stated results have been achieved on seen or unseen data. In general, the paper would benefit from a more in-depth analysis of the results: why does the proposed method perform so much better than the RBF-based GPs? Previous research has achieved convincing results without doing deep kernel learning. What is different about the task at hand in this paper that results in such a substantial drop in performance of the published baselines?

- Figure 3: replace the x-axis with time instead of sample count, add a grid on the plots.

- L204: at which rate is this data recorded? I.e. how much track expose time do these sample numbers correspond to?

- The size of the MLP used to do kernel learning has a very specific size. How was this network size found? What would happen if the network capacity increased/decreased? The learning rate of 0.02 is surprisingly high btw, wondering if this hyperparameter has been ablated or not.

- The writing of the paper could be improved, here is a list of the most prominent issues:
  - L43: minor stylistic feedback: a paper should not call its own method 'innovative'
  - L50: the method is not trained on the car, it's trained on data collected from the car
  - L266: an inference time should be reported in a unit of time, not a unit of frequency (i.e. 20-40ms inference time)


**Quality Of The Limitations Section:**

Limitations are addressed clearly

**Questions For Rebuttal:**

It would be interesting to understand if the superior performance of DKL-SKIP is across both the training and testing set or only on the testing set. I.e. is the model generalizing better or does it in general have better approximation capacity? The paper does not clearly state if the stated results have been achieved on seen or unseen data. In general, the paper would benefit from a more in-depth analysis of the results: why does the proposed method perform so much better than the RBF-based GPs? Previous research has achieved convincing results without doing deep kernel learning. What is different about the task at hand in this paper that results in such a substantial drop in performance of the published baselines?



**Robotics Focus:**

Relevant but unlikely to deploy to hardware in near future

**Summary Of Paper:**

The paper proposes a combination of GP regression with learned kernel functions for the task of residual dynamics learning for autonomous racing. The proposed method, termed DKL-SKIP, is shown to achieve better predictive performance on both real-world as well as simulated data.

**Summary Of Recommendation:**

My recommendation is based on :
- the promising results reported in the paper
- the missing interpretation of the results
- missing details about the evaluation procedure (are results obtained on seen or unseen data, sampling frequency, choice of hyperparamters)

---

### Comment · Area_Chair_yLxv · 2023-08-11
**Discussion until Aug 15th**

I would like to thank the authors and reviewers and would like to encourage you to make best use of the discussion period which will end *Aug 15 at 11:59 PM PT*.

In particular, to the reviewers: you are highly encouraged to engage in additional discussions with the authors if there are any remaining questions you would like the authors to elaborate on further.

---

### Decision · Program_Chairs · 2023-08-30

**Decision:**

Accept (Poster)

**Comment:**

The authors present an extended kinematic single track model for an autonomous race car. A GP model is used to learn the discrepancy between the extended kinematic model and observed dynamics in real and simulated environments. The proposed DKL-SKIP approach  is a deep kernel learning + SKIP-GP combination and is shown to outperform baselines in real and simulated experiments.

The reviewers recommendations where split with 3 weak accept and 1 strong reject recommendation.

Strengths of the proposed work include
- the paper was found to be well written
- valuable review of related work
- the inclusion of real world experiments
- relevance to autonomous driving community
- improved performance reported over state of the art

Areas of potential improvement:
- more challenging race tracks could be considered instead of oval tracks
- additional closed-loop evaluation was requested by a reviewer
- additional exploration and interpretation of the results was requested
- the degree of novelty of the approach was questioned and significance relative to existing
  and classical methods was an area of concern voiced
- additional critical reflection about the method was requested

In summary, I believe the work to be of interest to the CoRL community and the experimental evaluation is of value in particular,
but the lack of strong accept recommendations and mixed opinions by the reviewers and remaining concerns also indicate that it may be valuable to incorporate additional improvements. In particular, the inclusion of an additional focus on the qualitative interpretation of the results, its limitations etc could be strengthening the work further. This submission may therefore currently be just near the acceptance threshold.